# Food Practice Lifestyles: Identification and Implications for Energy Sustainability

**DOI:** 10.3390/ijerph19095638

**Published:** 2022-05-05

**Authors:** Leanne S. Giordono, June Flora, Chad Zanocco, Hilary Boudet

**Affiliations:** 1School of Public Policy, Oregon State University, Corvallis, OR 97331, USA; hilary.boudet@oregonstate.edu; 2Department of Civil and Environmental Engineering, Stanford University, Stanford, CA 94305, USA; jflora@stanford.edu (J.F.); czanocco@stanford.edu (C.Z.)

**Keywords:** food consumption, social practice, time use, energy sustainability

## Abstract

Food systems, including production, acquisition, preparation, and consumption, feature importantly in environmental sustainability, energy consumption and climate change. With predicted increases in food and water shortages associated with climate change, food-related lifestyle and behavioral changes are advocated as important mitigation and adaptation measures. Yet, reducing emissions from food systems is predicted to be one of our greatest challenges now and in the future. Traditional theories of environmental behavioral change often assume that individuals make “reasoned choices” that incorporate cost–benefit assessment, moral and normative concerns and affect/symbolic motives, yielding behavioral interventions that are often designed as informational or structural strategies. In contrast, some researchers recommend moving toward an approach that systematically examines the temporal organization of society with an eye toward understanding the patterns of social practices to better understand behaviors and develop more targeted and effective interventions. Our study follows on these recommendations with a study of food consumption “lifestyles” in the United States, using extant time use diary data from a nationally representative sample of Americans (*n* = 16,100) from 2014 to 2016. We use cluster analysis to identify unique groups based on temporal and locational eating patterns. We find evidence of six respondent clusters with distinct patterns of food consumption based on timing and location of eating, as well as individual and household characteristics. Factors associated with cluster membership include age, employment status, and marital status. We note the close connections between age and behaviors, suggesting that a life course scholarship approach may add valuable insight. Based on our findings, we identify opportunities for promoting sustainable energy use in the context of the transition to renewables, such as targeting energy-shifting and efficiency-improvement interventions based on group membership.

## 1. Introduction

Food systems, including production, acquisition, preparation, and consumption, feature importantly in environmental sustainability, energy consumption and climate change [1,2,3,4,5]. Recent estimates indicate that global food systems, including crop production, land use, livestock and fisheries, supply chain and diets, account for 21 to 37% of greenhouse gas (GHG) emissions [1,2,3]. The United States is among the top five producers of agri-food emissions, behind only China, India, Brazil and Indonesia [4]. Among total food system emissions, 8% are attributable to direct human consumption [5]. With predicted increases in food and water shortages associated with climate change, food-related lifestyle and behavioral changes are advocated as important mitigation and adaptation measures [6]. Yet, reducing emissions from food systems is predicted to be one of our greatest challenges now and in the future [5]. Indeed, moving toward energy sustainability, or “systems that use sustainable energy resources and that process, store, transport and utilize those resources sustainably” [7] p. 56, is a key element in efforts to address climate change. 

Achieving energy sustainability depends on multiple elements, both technical (e.g., harnessing renewable energy sources) and socioeconomic (e.g., modifying lifestyles) [7]. While we have made considerable technical progress, the second element—how to induce change in human attitudes and behavior—continues to pose challenges. Common theories of energy-focused behavioral change often assume that individuals make “reasoned choices” that incorporate cost–benefit assessment, moral and normative concerns and affect/symbolic motives, yielding behavioral interventions that are often designed as informational or structural strategies [8]. Utilities, for example, have long experimented with a variety of strategies to manage residential energy use—especially from peak to non-peak periods—including demand response, time-of-use pricing and promotion of “smart” energy technologies that engage energy users and automate selected functions. That said, Shove [9] broadly describes such approaches as the “ABC” paradigm—attitudes, behavior and choice—and suggests that they constrain our understanding of social change. She argues for a paradigm shift toward an approach that systematically examines the temporal organization of society with an eye toward understanding patterns of social practices as a means to designing truly effective programmatic and policy interventions [9,10]. These observations are echoed by Strenger’s call for increased attention to social practice in energy management [11] and Warde’s [12] broader proposition that “the source of changed behavior lies in the development of practices.” This study follows on these recommendations with a study of food practices in the context of social practice theory, which characterizes human behavior as coordinated patterns of action that are routinized and regularly reproduced [13,14]. 

We examine two research questions: (1) What are the temporal and locational patterns of food-related practice (i.e., food lifestyles) among the United States population? (2) What are the associations between these food lifestyles and individual and household characteristics? We use data from a 3 year panel (2014–2016) of the American Time Use Survey (ATUS) data, a nationally representative survey of Americans (*n* = 16,100) that includes both time use diary data and detailed information about individual and household characteristics. We find evidence of distinct food consumption lifestyles among the American public, characterized (on average) by clear “food consumption peaks” in the morning, midday and early evening. We also find evidence of intertemporal associations between food preparation, eating and cleaning, suggesting that these activities are often sequentially bundled. Finally, we find that food practices or activities vary considerably by location (at-home/not-home) and time of day, and that they are associated with unique individual and household characteristics including employment, household resources, age, gender and marital status. 

Our study makes several unique contributions to the existing literature. We advance practice-based approaches to energy demand with an in-depth examination of food-related practices both in and outside of the home using survey data from a representative sample. We identify lifestyles that are embedded in their individual and household characteristics, including life course stages and occupancy decisions. Finally, we respond to calls for more empirical evidence from practice-based studies, e.g., [15]. Our study is, to the best of our knowledge, the first detailed study that identifies distinct groups based on food consumption patterns via time use data in the American context.

## 2. Background

We turn to several literatures to inform our expectations and methods. First, we ground our research questions and our approach in social practice theory, as well as related empirical findings and connections to environmental and energy use decisions, e.g., [12,13,16,17,18,19]. We also draw inspiration from recent uses of nationally representative time use diary data in European settings, including the United Kingdom [20,21,22], France [23] and Denmark [24]. Literature on food consumption patterns in the United States informs our expectations about likely patterns and variations in food consumption lifestyles across a variety of individual and household characteristics, e.g., [25,26,27]. Finally, for motivation and methodological inspiration, we turn to the energy use and sustainability literature, especially the latter’s growing application of time use data—both of which use clustering techniques to characterize patterns of household electricity consumption—to explore individual and household activities, e.g., [20,28,29]. 

### 2.1. Social Practice Theory

Social practice theory broadly views human behavior (and changes therein) as a multiplicity of actions that tend to be coordinated, routinized and repeated regularly [12]. Descended from seminal work by Bourdieu [30,31] and Giddens [32], social practice theory stands in contrast to both individualist theories (e.g., rational choice theory) and structuralist theories (e.g., systems theory) [19]. Despite common origins, social practice theory is not a well-defined or unified theory of behavior. Practice theorists tend to view practices as arrays of activities (most often human activities), the combination of which form an interconnected field of practices, which are themselves comprised of various phenomena, including knowledge, language, and institutions. Importantly, a practice depends on the interconnectedness of various activities, and “is understandable to potential observers (at least within the same culture)” [13]. Beyond that point, social practice theory exhibits considerable diversity in its understanding of practices [19,33,34]. For the purposes of this study, we use a fairly limited definition of practice, identifying food consumption practice as arrays of activity that occur over time and space, to accommodate observations from other practice-based studies within the context of our objectives and extant (and somewhat inflexible) data source. In the aggregate, food-related “lifestyles” are comprised of the temporal practices embedded in food-related practices. 

The practice, a core concept of social practice theory, refers to the cultural conventions, sociotechnical systems, material objects, perceived norms, social and economic institutions, and spatial and temporal organization of relevant activities, specifically location, time of day, duration, sequence, periodicity, and presence of others [35]. Food-related practices clearly exhibit these characteristics, e.g., [34,36]. In addition to food consumption (i.e., eating), individuals and households engage in some amount of food purchase, preparation and cleanup. Almost all of these practices are likely to be dispersed over time, and to some degree, space. Food preparation activities tend to occur in the morning, afternoon and evening [20], for example, and are sequentially connected to the practice of eating due, at least in part, to the relationship between the outputs of cooking and the inputs of eating [37]. Eating and cleanup are similarly connected, with the outputs of eating (e.g., dirty dishes) serving as the input for cleanup, although the increasing engagement in simultaneous consumption activities may compress or even eliminate the sequential nature of food practice [12]. Food consumption practice also reflects social convention and the (frequently) social nature of meals [16,36]. Food practice further entails coordination and synchronization with other practices, such as social events or work [36,38]. Relatedly, Southerton [17] found that sociodemographic characteristics have a considerable influence on social practices, with age, gender, marital status and employment yielding distinct practices and degree of synchronization with other household members. With just a few exceptions, e.g., [20,39], past work in the social practice space has been largely qualitative, with small numbers of participants, thus limiting generalizability. 

There is considerable evidence that food-related practices frequently occur as sequences with a high level of time dependency that are synchronized among selected members of the household, but may change over the life course [16,17,20]. However, much existing research has focused on settings outside the United States. In a comparative analysis of five OECD countries from the 1970s to the 1990s, Warde et al. [40] found considerable persistence of national differences, suggesting that food practices in the United States are likely to differ from those observed in other international contexts. Despite some common trends, such as an overall decrease in the time spent eating at home, Americans spent considerably less time eating and preparing food than citizens of the other four countries [40].

### 2.2. American Food Consumption and Preparation Patterns

Scholarship on American eating and food preparation habits has tended to assess shifting trends, e.g., [41,42], and/or the aggregate time spent in various activities over the course of the day, e.g., [25,43], with little exploration of the temporal and locational nature of food-related practices. One notable exception is work by Flood et al. [44], which examined temporal patterns using a similar approach; however, that work examined all types of activities, mapped onto four analytic categories. In the United States, the late 20th century was characterized by a marked decline in home-based cooking and an increase in food consumed away from home, accompanied by high levels of secondary eating, or snacking while engaged in other activities [25,42,43]. Smith et al. [42] reported an increased share of daily energy intake consumed away from home, especially among higher-income groups, although these indicators were stable in the early 2000s, and estimated that approximately two-thirds of daily energy from food were consumed from home sources, with differences among income groups. While lower-income groups consumed a higher share of daily food energy from home sources and tended to spend more time cooking than higher-income groups, they nonetheless experienced a larger decline in the proportion who cook and the amount of time spent cooking during the last few decades of the 20th century, leading some analysts to conclude that they have shifted toward diets high in refined grains, fats and sugars [42]. 

Time use data have also revealed that a high share of Americans engage in secondary eating—defined in the ATUS as eating while engaged in other primary activities [45]—at least once during the day [25,26,27,43]. Secondary eating is not recorded if the respondent’s main activity is sleeping, primary eating/drinking, or eating/drinking as part of job. However, the implications for health outcomes are mixed. Bertrand and Schanzenbach [25] offered evidence that among American women, half of daily calories are consumed while engaged in secondary eating and drinking (i.e., while engaged in another non-eating activity), and suggested that secondary eating may be associated with obesity. In contrast, Hamermesh found that “grazing” (secondary eating/drinking) is skewed—most people do not graze or graze very little—and that grazing was more frequently associated with lower Body Mass Index (BMI) and better self-reported health [26]. Zeballos [43] suggests that while snacking occurs frequently, “snacking is generally fast”; and asserts that most calorie intake is likely to be captured by longer-duration primary eating events. 

Finally, some scholars have noted that time constraints may themselves contribute to distinct patterns of food practices. Kalenkoski [46] observed that the probability of experiencing “time poverty,” or not having enough discretionary time for personal self-care, health and human capital, increased considerably for employed individuals and for those in households with children. However, time poverty was not associated with income [46]. Subsequent research showed time poverty to be associated with different eating patterns, including lower likelihood of eating and drinking during the day, as well as lower likelihood of fast-food purchase [47]. Potential explanations for the somewhat counter-intuitive negative relationship between time poverty and fast-food consumption is the observation that fast-food “may not in fact be fast to those with limited time” [47] p. 98 and that time-poor individuals may purchase prepared foods from other venues (e.g., grocery stores).

### 2.3. Connections to Energy Use and Sustainability

There is considerable interest in understanding the potential for social practice (or changes therein) to either facilitate or hinder shifts in energy use, and sustainability more broadly. Food consumption practices and their patterns provide important insights for potential public policy interventions and strategies related to environmental and energy use behaviors. The timing of residential energy use is particularly important given the ongoing shift to renewable energy generation, which yields considerable fluctuations in the net load through the course of the day, commonly known as the “California duck curve” [48]. American energy used for household food storage and preparation accounts for approximately 28% of all energy consumed in the food production and consumption system [49]. Food consumption and related activities, especially those that involve electric appliances can influence individual- and household-level environmental impacts and energy consumption, as can other non-specific practice activities that support comfortable food practices, such as lighting and heating or cooling (HVAC). Similarly, food acquisition may involve purchasing choices (e.g., meat vs. non-meat; packaged vs. fresh foods) and transportation choices (e.g., public vs. private transportation; gasoline-vs. electric-powered vehicles). Finally, cleaning and disposal practices may also involve choices between more environmentally sustainable practices, such as recycling and composting, and less sustainable practices, such as direct disposal of food and packaging waste. Such household choices are often reflected in the timing and sequencing of food acquisition and consumption activities [45,50,51]. 

Moreover, while practice disruptions are tolerated and flexibility is considered normal [16], even flexible practices have their limitations. Recent scholarship from the United Kingdom, for example, found that food practices typically take place at the same time each day, with little seasonal variation [20]. Just as location-based constraints related to school and work, for example, limit flexibility in energy use [52], highly synchronized (e.g., serving hot food) or culturally grounded practices (e.g., taking tea) may also be relatively fixed [38,53]. Based on these observations, we suspect that occupancy patterns (i.e., being at home or away from home) and other demographic characteristics will be reflected in individuals’ food-related lifestyles. Some inflexibility may also be related to the phenomenon of time poverty [46,47]. While little scholarship has attended to the flexibility of food practices in the United States, Stelmach et al. [39] made relevant observations, finding that households may be less willing to shift some home-based activities that occur at peak hours, such as cooking dinner, while willing to shift others, such as running the dishwasher. A better understanding of food practices has the potential to yield insights into daily American energy and environmental behaviors. 

## 3. Methods

### 3.1. Data

As our primary data source, we used extant American Time Use Survey (ATUS) data collected by the US Census Bureau [54]. All ATUS data were accessed via IPUMS, a database of integrated and documented Census and survey data [55]. ATUS survey respondents were selected randomly from households that completed an 8-month rotation of the Current Population Survey (CPS). The households in the CPS sample (from which ATUS sample was drawn) were stratified based on race/ethnicity of household head, presence and age of children. Households with a Hispanic or non-Hispanic Black household head and households with children were over-sampled. Respondents were limited to individuals over age 14. The sampling processes yielded a nationally representative sample of US households. See Appendix A for more detailed information about the ATUS sample.

The ATUS interview collected information on all activities conducted during a 24 h period that began at 4:00 a.m. on a designated day and ended at 3:59 a.m. on the following day. Specifically, the interviewer asked each respondent the following question: “Yesterday, [previous weekday] at 4:00 a.m., what were you doing? What did you do next?” Each respondent listed his/her primary activities in order, describing each activity in which he/she was engaged, as well as the location where the activity occurred. Information on the activities were recorded sequentially with ending times, from which the survey instrument calculates start time and duration. In the final data set, coded activities were represented using a 3-tier, 6-digit coding structure. 

The standard ATUS interview only records information on primary activities, with the exception of childcare, which was recorded as a secondary activity. The ATUS Eating and Health Module, however, collected additional information on eating as a secondary activity, as well as multiple supplement variables related to eating and nutrition. The Eating and Health module was only available for the 2006–2008 and 2014–2016 ATUS data; given our interest in food-related activities, we elected to use the 2014–2016 data in order to take advantage of this module. We restricted our analytic sample to include only respondents who responded with an account of their activities during a weekday (i.e., Monday through Friday) and who responded to the Health and Eating Module. Our final analytic respondent sample included 16,100 individual-level observations. See Appendix A for a detailed description of our analytic sample.

Distinct primary food-related activities that were captured by the ATUS and the Eating and Health module included eating and drinking; eating and drinking as a part of job; travel associated with eating and drinking; food and drink preparation; kitchen and food cleanup; grocery shopping; purchasing food (not groceries); traveling to/from the grocery store; and travel related to purchasing food. With the exception of sleeping, grooming, and personal activities, the ATUS gathered detailed information about the location where each primary activity was conducted. The Eating and Health module also included information about secondary eating, which measures the time spent eating during other primary activities; secondary eating was not recorded if the primary activity was sleeping, primary eating/drinking or eating/drinking as part of job. Eating, drinking and caregiving (childcare and elder care) were the only secondary activities captured by the ATUS. Eating and drinking were recorded together when captured as a primary activity. In contrast, eating and drinking were recorded as separate secondary activities; for the purpose of this analysis, we have only used secondary eating (not secondary drinking). For the purpose of this study, we use the term “eating” to indicate eating and drinking as a primary activity, plus eating (only) as a secondary activity; the ATUS does not distinguish between eating and drinking when measured as a primary activity. See Appendix A for a list of variables used in our analyses. 

The data were originally downloaded as a rectangular data file. Activities of interest were subsequently reshaped and collapsed to form multiple analytic files that included binary (yes/no) and continuous (duration) indicators for each activity of interest for all 30 min time intervals during the course of the 24 h survey period. Finally, we linked activity data to a variety of relevant CPS individual and household characteristics, including demographics; education, employment and marital status; household size, income, poverty status, presence of children and metro status; and recent fast-food consumption. Individual0 and household-level variables and categories used in our analyses, with IPUMS variable name, label and original format, are provided in Appendix A. More documentation about the ATUS interview, including detailed sampling information and interview items, can be found at the IPUMS online platform [55].

### 3.2. Analytic Approach

We used a combination of descriptive statistics, cluster analysis and multivariate analysis approaches to address our research questions of interest. For all respondents, we initially aggregated activity-level data into the duration of time spent in the activity of interest per 30 min interval (4:00–4:30 a.m., 4:30–5:00 a.m., etc.), separated by location (at-home or not-home). Stata 15.1 (StataCorp, College Station, TX, USA) was used to perform all analyses. 

The ATUS method of time diary recording yields inconsistent intervals across participants, depending on the start and end times of each activity. To facilitate analysis, we began by identifying the number of minutes spent in each activity over 48 non-overlapping intervals of 30 minutes each (*n* = 772,800). The data in these intervals were used to generate descriptive statistics and related visuals for each interval, including the mean duration (minutes) spent in each activity and interval, as well as the proportion of the population reporting each activity during the interval. After aggregating duration and binary participation data by person and location-specific activity, we generated a series of descriptive statistics for the sample as a whole and described the extent of participation in the activities of interest by 30 min interval. 

Specifically, we examined four key elements of food practice: prevalence, duration, periodicity and sequential bundling, as shown in Table 1. We define *prevalence* as the proportion of the population who engage in various food-related activities. We define *duration* as the total time spent in food-related practices, both as a percentage of the day and as a percentage of time spent in all food-related activities. *Periodicity* is defined as the pattern of events over the course of the day, while *sequential bundling* reflects the order and combinations in which selected food-related activities occur. We examined each of these elements for two main locations—home and away-from-home—and in combination. 

We used a series of descriptive statistics to explore prevalence and duration and periodicity. We used a fixed-effects time series model with robust standard errors to examine the associations between at-home eating in any given 30 min interval, and food preparation and cleanup in any given period and in the two periods before and after. (For this analysis, we aggregated secondary eating/drinking with primary eating/drinking when the number of minutes spent in secondary eating during the interval exceeded the number of minutes spent in the primary activity. In some analyses, we separated secondary eating from primary eating.) For all descriptive statistics, we applied ATUS probability weights. Note that standard Stata panel analysis commands (xtreg and xtlogit) preclude the use of probability weights, so they were not applied in the time series model.

We then employed a multistep K-means cluster analysis method to identify groups of individuals exhibiting distinct eating patterns over the course of 30 min intervals during a 24 h period. K-means cluster analysis partitions observations into clusters in where each observation belongs to the cluster with the nearest mean [56] and offers a non-parametric method for identifying discrete groups—in this case, groups of respondents with similar food “lifestyles.” The choice of the k-means for clustering individuals was informed by both practical considerations and for its comparability to related fields examining lifestyles. First, the k-means algorithm allows us to construct a mean shape of eating patterns across a 24 h period for each cluster, as resultant cluster centers represent the mean time spent eating during each time interval for all of the individuals within the cluster. This allows us to capture information not only about the magnitude of eating activities but when these activities occurred during the day. Second, this approach to clustering daily time series data with hourly or sub-hourly intervals is similar to that employed in segmenting residential energy consumption (i.e., load shape) profiles into groups of households using smart meter data [29]. This allows us to easily identify different patterns in food activities across clusters using representative food activity time-based profiles. Finally, clustering has also been used in applications for lifestyle segmentation using residential electricity consumption data to identify households for enrollment certain energy programs or interventions [57].

The cluster analyses were performed on the normalized duration of time spent eating over 41 overlapping 4 h periods. The duration of each 4 h period reflects the sum of eight sequential 30 min intervals, which were combined to overcome issues related to data scarcity. Each 4 h period was normalized by the total amount of time spent eating/drinking (including secondary eating) during the course of the 24 h period so that cluster identification would reflect the overall shape of the cluster, rather than the total duration spent eating. 

Finally, we used descriptive statistics and multivariate regression to describe the clusters and predict cluster membership. We modeled cluster membership using multivariate logistic regressions performed separately for each of the largest six cluster combinations (i.e., cluster composed of at least 2% of the weighted sample). 

## 4. Results

### 4.1. Temporal and Locational Patterns of Food-Related Practice

#### 4.1.1. Prevalence

Almost all respondents (99%) reported eating at some point during the day, with most engaging in at-home eating (88%) and a slightly smaller share engaging in eating away from home (61%). A little over half (54%) reported secondary eating. Other activities were somewhat less prevalent in the population. Just over half of respondents engaged in food preparation (55%), while less than one-quarter participated in cleanup activities (24%); the vast share of those activities occurred in the home. See Appendix A for results by location.

#### 4.1.2. Duration

As shown in Figure 1, time spent in food-related activities also varied considerably. On average, respondents spent a little over an hour eating (79 min, or 5% of the day). Among those who ate at home, the average number of minutes spent eating was about the same as those who ate away from home (53 and 52 min, respectively; 4% of the day for each). Among the activities that mostly took place away from the home, grocery shopping was the longest-duration activity (43 min; 3%) of the day, followed by food-related travel (25 min; 2% of the day) and other food purchase (11 min; 1% of the day). See Appendix A for detailed conditional and unconditional estimates. Our analyses also revealed that considerably more time was devoted to primary eating (65 min; 5% of the day) than secondary eating (30 min; 2% of the day). Not shown in figure.

#### 4.1.3. Periodicity

We then examined the periodicity of food-related activities. As shown in Figure 2, eating remained the highest-prevalence activity throughout the day, followed by food preparation, both of which exhibit a clear 3-peak pattern in the morning, midday and evening. To facilitate interpretation, we use four broadly defined periods of the day, including morning (4:00 a.m. to 10:00 a.m.), midday (10:00 a.m. to 4:00 p.m.), early evening (4:00 p.m. to 9:00 p.m.) and night (9:00 p.m. to 4:00 a.m.) After a morning peak of 14% at 8:00 a.m., the highest peak occurred at midday, with over one-quarter (26%) eating lunch from 12:00 to 12:30, followed closely by another evening peak (24% from 6:00 to 6:30 p.m.). Food preparation also showed three distinct peaks with lower prevalence than eating, with the highest food preparation peak occurring in the early evening during the 5:00–5:30 p.m. interval (11%). This evening food preparation peak occurred prior to the evening eating peak. Other food-related activities showed much lower prevalence throughout the day, with slight peaks for food cleanup in the early evening (4% from 6:30 to 7:00 p.m.) and food-related traveling (3% from 12:00 to 12:30 p.m.). When examined by location (also Figure 2), we note that eating was divided between home and away, while food preparation and cleanup primarily in the home, and grocery shopping, other food purchase and food-related travel occurred outside the home. The figure also demonstrates that the midday eating peak away from home (16% from 12:00 to 12:30 p.m.) exceeded the at-home midday peak (10% from 12:00 to 12:30 p.m.), while the at-home early evening peak (19% from 6:00 to 6:30 p.m.) exceeded the away-from-home peak (6% from 6:00 to 6:30 p.m.).

We also explored the duration patterns of food-related activities over the course of the day, by primary/secondary eating patterns, and by selected food-related activities. The results from these analyses echo our results from above, namely indicating a 3-peak eating pattern with distinct patterns by location. We also found that overall, secondary eating does not exceed 1% of all time spent eating in any interval, with slight differences by location. See Appendix A for detailed results.

#### 4.1.4. Sequential Bundling

We expected that for some time periods, the occurrence of eating during the interval was more likely to be preceded by food preparation and followed by cleaning, representing activity bundling. Given earlier results for cleaning and preparation, which indicate that these activities primarily occur in the home, we only examine sequential bundling at home. The results shown in Figure 3 offer evidence of sequential bundling, with positive and statistically significant relationships between eating, pre-eating food preparation (up to two periods prior) and post-eating cleaning (up to one period after). See Appendix A for regression tables.

### 4.2. Clustered Patterns of Food Practice

#### 4.2.1. Eating Patterns by Cluster

Our second research question asks what individual or household characteristics are associated with distinct eating patterns, and relatedly, food “lifestyles”. The cluster analysis, conducted separately on at-home and not-home eating/drinking, identified three clusters for each location. We used a combinatorial approach to assign cluster membership to each respondent based on unique combinations of at-home and not-home cluster memberships. Specifically, we used the combinations of three clusters identified in each location to create nine clusters. Table 2 illustrates our overall combinatorial approach; in this approach an individual with membership in At-Home Cluster 1 and Not-Home Cluster 2 would be assigned to Cluster 4, while an individual with membership in At-Home Cluster 3 and Not-Home Cluster 2 would be assigned to Cluster 6. Ultimately, we retained the six largest clusters, representing over 98% of the respondent sample. We excluded respondents from the smallest three clusters (who comprised less than 2% of the sample) from our description of the results. See Appendix A for additional details about the cluster analysis process.

#### 4.2.2. Eating Patterns by Cluster

As shown in Figure 4, each cluster exhibited a distinct combination of temporal and locational eating patterns, as a proportion of time spent in selected food-related activities (preparation, eating and cleaning). For convenience, we have named the clusters: (1) Home for Dinner (24% of the sample); (2) Home for Breakfast, Lunch and Dinner (23%); (3) Out for Lunch (20%); (4) Home for Breakfast and Dinner (12%); (5) Out for Dinner (11%); (6) Out for Lunch, Home for Dinner (8%). In selected figures and descriptions, we abbreviate the meal as follows: B = Breakfast, L = Lunch, and D = Dinner. 

#### 4.2.3. Cluster Characteristics

In addition to exhibiting distinct eating patterns, the clusters can be easily distinguished on the basis of individual and household characteristics. Table 3 presents cross-tabulations by cluster and demographic variables of interest, followed by detailed descriptions of each cluster. Results from additional analyses, including participation in other food-related activities and activity duration, are shown in Appendix A, while the results from multivariate analyses are provided in Appendix A.

##### Cluster 1: Home for Dinner (24%)

The largest group (Cluster 1, 24%)—the “Home for Dinner” cluster—represents individuals for whom a considerable share of total eating time was spent at home in the evening, although this cluster was also characterized by the fewest total minutes spent eating (mean—65). Fairly high shares of cluster members engaged in both food preparation (62%) and cleaning (29%). On average, cluster members spent a relatively long amount of time in food preparation (57 min) and cleanup (35 min). Compared to the overall sample, a relatively high share of this cluster were married (57%), female (54%) and lived in households with 3 or more individuals (27%), and a smaller proportion were employed (55%). Multivariate results indicated that the odds of belonging to the cluster were higher for married respondents, and lower for respondents who were employed, age 65+, non-Hispanic, and respondents who reported having eaten fast food in the previous week.

##### Cluster 2: Home for Breakfast, Lunch and Dinner (22%)

Cluster 2 (23%), or the “Home for Breakfast, Lunch and Dinner” cluster, was composed of individuals who displayed home-based eating throughout the day, with a midday peak. The cluster displayed the largest share of individuals engaged in preparation (65%) and cleaning (33%), and also spent the longest average time in preparation (62 min) and cleaning (39 min). This cluster included the largest share of older adults aged 65+ (33%), respondents who were not in the labor force (57%), and respondents in near-poverty households (42%). When controlling for multiple variables, we found that the odds of cluster membership were higher for adults age 65+, females, respondents in households ≤185% of poverty, and lower for employed respondents, young adults age 15–24 and those who reported eating fast food.

##### Cluster 3: Out for Lunch (20%)

The “Out for Lunch” cluster—individuals in Cluster 3 (20%)—showed the highest share of eating time at work with a midday peak, and only a small peak in the evening at home. Less than half (48%) of the cluster spent any time in food preparation activities, and only 16% spent time in food cleaning. The “Out for Lunch” cluster was comprised of a high share of employed (81%) and working-age adults (72%) and the largest share of Hispanic respondents (20%), all of which corresponded with significant predictors in the multivariate analysis. A fairly high proportion of this group reported purchasing fast food in the prior week (65%). Our multivariate results also indicate that the odds of membership in the cluster were higher among individuals who report having consumed fast food, and lower among individuals in near-poverty households and among married individuals.

##### Cluster 4: Home for Breakfast and Dinner (12%)

Cluster 4 (12%)—“Home for Breakfast and Dinner”—was composed of those for whom morning and evening eating represented the bulk of total eating, which tended to occur at home more often. Compared to other clusters, Cluster 4 includes the lowest share of respondents (84%) who reported having engaged in primary eating, and the highest share (58%) who reported secondary eating. Members of the cluster showed the lowest engagement in preparation (39%) and cleaning (18%). Cluster 4 respondents were more likely to be male (56%) and Black (17%), and less likely to be married (47%). Our multivariate analysis indicates that the odds of membership in this cluster were higher among employed individuals, but lower among young adults, females, Whites, and married respondents.

##### Cluster 5: Out for Dinner (11%)

The “Out for Dinner” cluster, or Cluster 5 (11%), spent a high proportion of their time eating away from home in the evening, and spent the highest average amount of total time engaged in eating (99 min). They were also characterized by the highest reported participation in secondary eating (59%). In contrast, cluster members showed the lowest participation in preparation (39%) and cleaning (12%) and spent the least amount of time in those activities (33 min and 24 min, respectively). They spent the most time eating relative to other food-related activities. Members of the “Out to Dinner” cluster were more likely to be young adults aged 15–24 (26%), living in single- or two-person households (78%) and experience a low likelihood of being in near-poverty status (25%). We also noted that members of Cluster 5 were more likely to have been interviewed about Friday activities (29%) and were more likely to have been interviewed in the summer months of June–August (30%). These variables are also significant predictors in the multivariate regression. This group also reported the highest share of fast-food purchase (67%). The odds of membership were higher among young, employed and highly educated respondents, as well as those who reported eating fast food. Odds of cluster membership were lower among Hispanics, married respondents and those with a young child in the household.

##### Cluster 6: Out for Lunch and Home for Dinner (8%)

Members of Cluster 6 (8%)—the “Out to Lunch and Home for Dinner” cluster—exhibited dual peaks, with one midday peak away from home and a later evening peak at home. Similar to Cluster 1, they showed considerable participation in preparation (58%) and cleaning (26%), although they spent less time on both activities than other clusters (35 min and 27 min, respectively). That cluster tended to be employed (85%), married (62%), working-age adults (77%) with young children (18%) and high educational attainment (BA or higher, 36%). Employment was an important predictor of membership in Cluster 6, with odds of membership three times those of unemployed respondents. Young adults, Whites, married respondents and those with young children also had higher odds of membership in Cluster 6, while the odds of membership were significantly lower for older adults.

## 5. Discussion

Our study highlights several American food lifestyles patterns, including evidence of a 3-peak eating pattern, distinctions between at-home and out-of-home food consumption, and the bundled nature of food practices. These observations echo recommendations by Blue et al. [35] to conceptualize flexibility as an outcome of social practice. Moreover, while eating occurred in both locations, food preparation and cleanup was largely reserved for home, and grocery shopping, food purchase and food travel primarily occurred outside the home. While these activities occurred outside the home somewhat by definition, changes in online shopping and food purchasing options (e.g., online shopping and food delivery), as well as work habits (e.g., working from home) could shift the location of these activities for some segments of the population. 

Perhaps more notable are our observations about variation in food-related patterns based on individual and household characteristics. Our combinatorial cluster analysis distinguished respondents based on both at-home and not-home food consumption patterns. We found evidence of six major respondent groups: (1) Home for Dinner (24% of respondents); (2) Home for Breakfast, Lunch and Dinner (23%); (3) Out for Lunch (20%); (4) Home for Breakfast and Dinner (12%); (5) Out for Dinner (11%); (6) Out for Lunch, Home for Dinner (8%). Four of these groups, representing over two-thirds of the sample, displayed clear at-home eating peaks. In contrast, one-third of the sample—“Out for Lunch” (20%) and “Out for Dinner” (11%)—did not show evidence of at-home eating peaks. Moreover, while the overall population was characterized by a 3-peak pattern (on average), we found that the groups’ patterns tended toward 1- or 2-peak cycles, with peaks at different times of day and in varying locations. These clusters also exhibited considerable variation in other food-related activity patterns, and were identifiable by key demographic variables. For example, “Home for Breakfast, Lunch and Dinner” appears to be dominated by older, retired adults who spend considerable time preparing, eating and cleaning at home throughout the course of the day. In contrast, the “Out for Lunch, Home for Dinner” cluster appears to include working-age adults with young children who are working away from home during the day but home for dinner at night. Additionally, the “Out for Dinner” cluster seems to consist primarily of young, single, and employed adults. 

These observations suggest a clear connection between food “lifestyles” and a life course scholarship approach, which explicitly acknowledges the close connections between age and behaviors, and which has been used in other contexts (e.g., social insurance) to identify policy interventions, e.g., [58,59,60]. Finally, we found positive associations between eating and other food-related activities in previous intervals. Taken together, our observations suggest that efforts to shift practices may be more practical among targeted groups who are comparatively time rich or exhibit other characteristics associated with distinct food-related practices likely to accommodate change. There may also be opportunities to target messaging about the environmental impacts of specific activities, such as consumption of highly processed food, among selected groups, such as those with a high level of fast-food consumption.

Our findings are subject to a number of limitations. First, by turning to the ATUS, we are limited to an examination of cross-sectional data that reflect the experience of individual respondents over the course of one day. We cannot validate the stability of the observed patterns for individuals over time, nor can we extrapolate our conclusions beyond the level of the individual (e.g., to household-level practices). Lastly, our approach (to some degree) skirts some of the main tenets and elements of social practice theory by reducing the complex and social nature of practices to a parsimonious analytic approach. These limitations highlight opportunities for future research. For example, we intentionally limited our analysis to weekdays. We suspect that deeper examination of differences over the course of the week, or a similar analysis of weekend food practices, would be fruitful. We also acknowledge that some of the activities examined in this study, such as grocery shopping and traveling for food, may not occur on a daily basis and may require an examination of an alternative interval (e.g., weekly) to understand the implication for food consumption practices. Finally, we recognize that a longitudinal and/or mixed-methods approach would likely yield richer information about food lifestyles, including stability over time and within households. 

While our study does not directly demonstrate a link between food lifestyles and environmental sustainability, our research points to potential applications using various clusters’ responsiveness to public policy interventions that promote sustainable actions. For example, while this study does not directly examine associations between food practices and energy use, insights from temporal studies of such connections are relevant for consideration. Consumer behavior and appliance efficiency both offer important opportunities for energy conservation [61]. Two main dimensions of energy reduction—curtailment and efficiency improvements—have been shown to be associated with distinct predictors [62]. Whereas efficiency improvements tend to be driven by several demographic characteristics, including age, gender, marital status and homeownership, as well as bill consciousness, curtailment is mostly explained by home occupancy, as well as multiple psychographic variables, including environmental concerns and motivation, as well as bill consciousness [62]. Our findings yield additional information about how different groups of Americans might respond to interventions intended to influence the timing of residential energy use, such as time-of-use electricity rate pricing. Yet, the potential for food practices to shift remains unexplored, yielding an interesting area for future research. Stelmach [39] suggests that practice flexibility may be linked to specific actions, based on a household being unwilling to shift some activities that occur at peak hours, such as cooking dinner, while willing to shift others, such as running the dishwasher. This study validates our expectations that food-related patterns are likely to exhibit a systematic pattern that varies across individuals and highlights the close intertemporal connection of home-based food preparation, eating and cleanup. Our findings also suggest that more refined targeting and messaging may improve the effectiveness of ongoing energy-shifting efforts. Some groups, such as the “Home for Breakfast, Lunch and Dinner” cluster, may be more amenable to efforts to shift their energy use away from peak windows, either because they are more likely to be present in the home during non-peak times or because they face budget constraints that would make them more responsive to time-of-use pricing. Other groups who are likely to be away from home in the evening, such as the “Out to Dinner” cluster, may not need the same level of targeting with respect to energy-shifting programs. Still other groups, such as the “Out to Lunch, Home for Dinner” cluster, may be less likely to respond to energy-shifting messages, but more responsive to efficiency-improvement interventions. We also foresee opportunities to develop targeted public education about food-related transportation costs. For example, by highlighting the consumer and societal energy-related costs of both consumer travel and food choices among Americans who regularly eat meals away from home, especially when such meals take the form of packaged foods. Opportunities to explore more directly the connections between these patterns of food-related activities among Americans and residential energy use would go far to advance both theory and practice.

## 6. Conclusions

Understanding such food practice dynamics is particularly important as households in the U.S. and across the world have dramatically shifted residential occupancy patterns and at home activities due to the COVID-19 pandemic [63], with these changes in household patterns having implications for electricity consumption and greenhouse gas emissions [64,65]. Additionally, while many areas across the world are beginning to return to some degree of normalcy, many of the changes to household routines, practices, and other elements of lifestyles may persist into the future. In this respect, the food practice lifestyle analysis approach that we introduce in this study is well suited for understanding not only how food-related patterns may have changed, but how these changes, in the aggregate, could impact electricity consumption, suggesting its usefulness for informing interventions and policies designed to better align the timing of household electricity consumption patterns with grid-level electricity production. This alignment of electricity demand and supply will only become more critical as we transition to a decarbonized electrical grid with a high penetration of renewable energy resources and joins a growing suite of demand-side solutions for climate change mitigation [66]. 

This study was primarily motivated by the potential implications of food lifestyles for sustainability, especially energy sustainability. Future research would examine these connections more directly. However, we also suspect that our approach is applicable to other social practices, such as personal care and caregiving, and that our findings offer insights into other areas of interest, such as public health. For example, our observations might inform hypotheses about the connection between food lifestyles and nutrition. As an alternative to the ABC paradigm—the study of attitudes, behaviors and choices—the social practice approach offers an opportunity to explore how practices are ordered across time and space, with applicability to a wide variety of contemporary public policy problems. 

## Figures and Tables

**Figure 1 ijerph-19-05638-f001:**
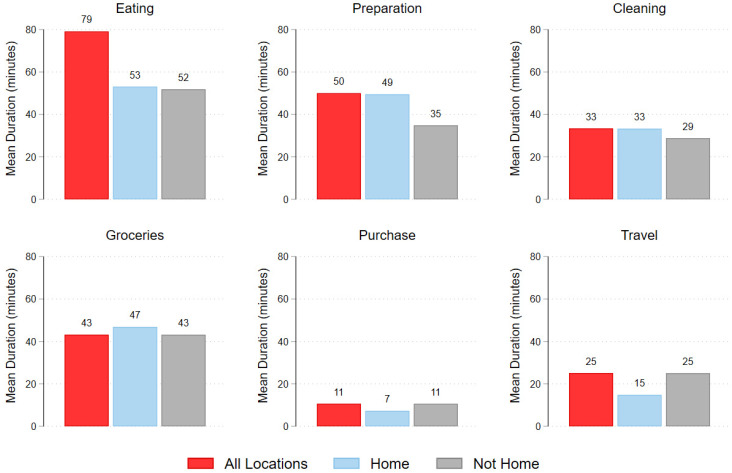
Mean duration (minutes) for food-related activities.

**Figure 2 ijerph-19-05638-f002:**
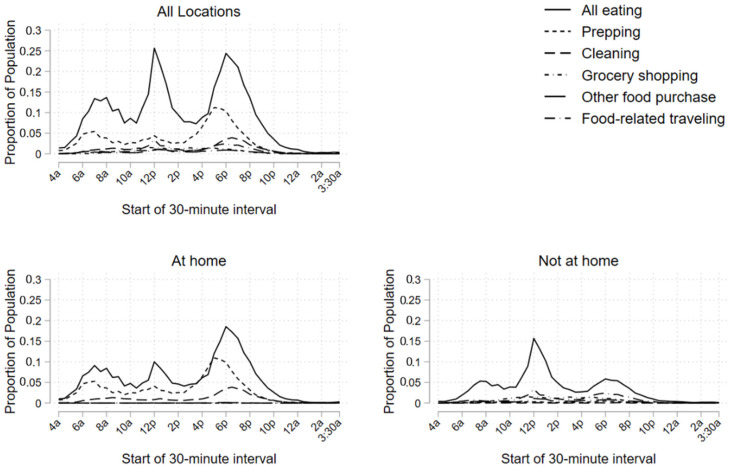
Proportion of population participating in all food-related activities by interval and location.

**Figure 3 ijerph-19-05638-f003:**
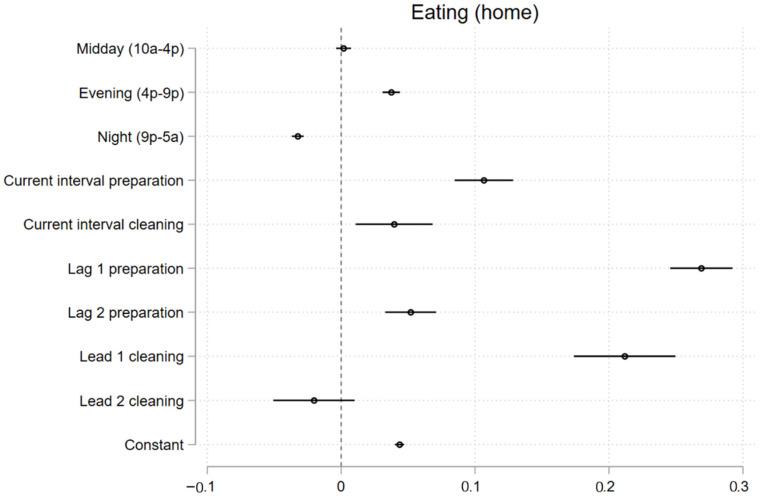
Multivariate panel regression predicting eating at home.

**Figure 4 ijerph-19-05638-f004:**
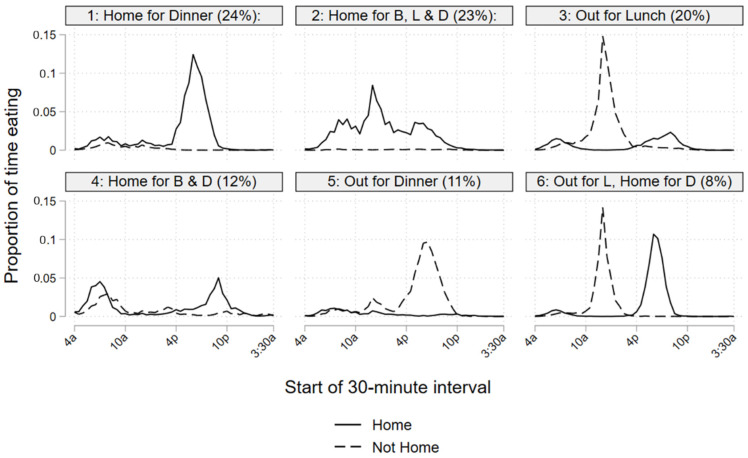
Time spent in primary and second eating by interval and location (as proportion of all eating).

**Table 1 ijerph-19-05638-t001:** Key Food Practice Elements.

Food Practice Element	Definition
Prevalence	Proportion of population engaging in various food-related activities
Duration	Duration of all food-related activities as (1) total time spent; (2) a percentage of the day; and (3) a percentage of all food-related activities
Periodicity	Pattern of events over the course of the day, including peak duration at various intervals (morning, midday, evening)
Sequential Bundling	Sequence of selected food-related activities (preparation, eating/drinking, cleaning) in combinations that occur in close temporal and locational proximity

**Table 2 ijerph-19-05638-t002:** Combinatorial Approach to Cluster Membership.

		At-Home Cluster Membership		
		Home Cluster 1	Home Cluster 2	Home Cluster 3
**Not-Home Cluster Membership**	**Not-home** **Cluster 1**	Cluster 1	Cluster 2	Cluster 3
	**Not-home** **Cluster 2**	Cluster 4	Cluster 5	Cluster 6
	**Not-home** **Cluster 3**	Cluster 7	Cluster 8	Cluster 9

Table shows potential combinations of at-home and not-home cluster membership.

**Table 3 ijerph-19-05638-t003:** Descriptive Statistics for Sample and by Cluster (*n* = 15,872; survey weights applied).

		Percent ^2^						
Variable	Category	Sample ^1^	(1) Home Dinner	(2) Home B, L and D	(3) Out to Lunch	(4) Home B and D	(5) Out to Dinner	(6) Out L, Home D
Age	15–24	16.7	15.8	12.8	18.4	15.0	25.7	16.1
	25–64	65.2	65.5	54.1	71.9	69.8	61.9	76.5
	65+	18.2	18.7	33.1	9.7	15.2	12.4	7.4
Sex	Male	48.4	46.0	42.1	52.8	55.9	48.1	51.4
	Female	51.6	54.0	57.9	47.2	44.1	51.9	48.6
Race	White	80.9	80.9	80.9	80.9	77.1	81.8	85.3
	Black	12.3	12.2	12.7	12.1	16.8	10.6	7.3
	Asian	4.3	4.0	4.4	4.6	3.9	4.3	5.2
	NA/PI/ HI	1.1	1.3	0.8	0.9	1.0	1.2	1.4
	Multiple	1.4	1.6	1.2	1.5	1.3	2.0	0.8
Ethnicity	Hispanic	15.9	14.1	16.7	19.8	15.2	12.3	15.5
	Not Hispanic	84.1	85.9	83.3	80.2	84.8	87.7	84.5
Education	No BA	44.1	45.8	49.1	41.2	44.1	37.4	41.1
	BA and higher	25.4	25.5	24.8	25.0	26.3	28.0	22.7
		30.5	28.7	26.1	33.7	29.5	34.6	36.2
Employment	Employed	61.7	54.8	36.2	80.6	67.7	73.0	84.9
	Unemployed/NILF	38.3	38.2	56.9	17.1	26.9	22.8	13.8
Marital Status	Married	52.2	56.9	54.0	49.0	47.0	42.9	61.9
	Not married	47.8	43.1	46.0	51.0	53.0	57.1	38.1
Household Size	1–2	75.3	72.7	80.0	73.6	75.8	77.9	66.6
	3–5	14.3	15.6	11.4	14.8	14.0	12.7	21.6
	6+	10.0	11.3	7.9	11.3	10.1	9.4	9.7
Children < 5 in HH	No child(ren) < 5	86.6	85.8	86.9	86.9	87.3	89.5	81.9
	Child(ren) < 5	13.4	14.2	13.1	13.1	12.7	10.5	18.1
Home Ownership	Owned	72.8	73.5	73.7	70.6	70.1	72.8	77.0
	Rented	27.2	26.5	26.3	29.4	29.9	27.2	23.0
Family Income	<25 K	23.2	23.9	32.7	17.6	25.3	17.4	13.3
	25 K–49 K	22.4	22.8	25.4	21.0	22.3	18.9	21.4
	50 K–99 K	27.0	26.8	22.5	29.3	26.3	31.5	29.7
	100 K+	27.3	26.5	19.3	32.2	26.1	32.2	35.6
Household Poverty	Income ≥ 185%	65.2	63.7	54.0	72.4	64.6	73.5	74.3
	Income < 185%	32.2	33.3	42.2	25.8	33.0	24.9	24.5
	Ref/DK/NIU	2.6	3.1	3.7	1.8	2.4	1.6	1.3
Purchased fast food	Did not purchase	42.9	47.0	52.2	35.2	42.5	32.4	37.3
	Purchased	56.8	52.5	47.4	64.5	57.3	67.0	62.4
	Refused/ DK/NIU	0.4	0.5	0.4	0.3	0.2	0.6	0.3
Metro Status	Metro	84.0	84.0	82.1	86.0	82.9	85.9	83.5
	Nonmetro	15.2	15.1	16.8	13.6	16.0	13.7	15.7
	Metro NA	0.8	0.9	1.1	0.4	1.1	0.4	0.8

^1^ Sample excludes members of the three smallest clusters. ^2^ Pearson chi-square tests of independence are significant at the *p* < 0.001 level for all variables.

## Data Availability

All data are available at the Census Burea’s ATUS website (https://www.bls.gov/tus/home.htm) or at the IPUMS website (https://www.ipums.org).

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
