# Peer review of "Food Practice Lifestyles: Identification and Implications for Energy Sustainability"

_ijerph, 2022, doi:10.3390/ijerph19095638_

Round 1

Reviewer 1 Report

This is an interesting study to use a time-use data for food-related lifestyles in the United States. The authors may consider changing the title because the research failed to clearly show how 'environmental sustainability' is related to their analysis results. In addition, the introduction and background are quite long and less relevant to the method section. The use of 'spatial' may be also misleading because the analysis shows only home vs. non-home. Since apparently there is a volume of studies using the same data in the U.S., it would be helpful for readers to identify this study's uniqueness if the authors can provide more information.

Author Response

Thank you very much for your comments. Please see attached Word document for our detailed response.

Reviewer 2 Report

I would like to thank the author(s) for your submission and appreciate the opportunity to read and review your manuscript. The study conducted cluster analysis to identify unique groups based on temporal and spatial eating patterns.

The writing of the introduction was organized well, and research and clinical implications were well addressed.

Could you provide the rationale why the author(s) used k-mean cluster analysis? Suggested using advanced statistical methods, such as latent class analysis.

In addition, the author(s) labeled 6 clusters: 1) Home for Dinner, 2) Home for Breakfast, Lunch and Dinner, 3) Out for Lunch, 4) Home for Breakfast and Dinner, 5) Out for Dinner, and 6) Out for Lunch, Home for Dinner. Not clear whether each reflects appropriately on each cluster’s characteristics. Please consider re-naming the clusters.

Author Response

(The authors gave the same response as above.)

Reviewer 3 Report

Nice piece of research to read about and very worthy of bringing to the forefront of future research and intervention priorities.

Abstract
The aim and research questions and how they extend on to the ATUS data could be clearer in the abstract and presented prior to the methods sentence. "We use...".  Consider including the interview numbers (n=16,100) and observations (n=772,800) for the reader upfront.

Introduction

Consider finishing the section when you clearly state your two research questions.

Line 96 - "We use... to address our research questions." should be in the method analyses section. 

Likewise, Line 100 "We find evidence of distinct food consumption...." These should be in results section.

Line 106 would be better placed at start of Background section.

Line 116 would be better placed in Discussion.  It was a little confusing to ascertain if all this detail is previous findings or the study you are presenting when it is upfront like this.

Line 195 - "Subsequent research...during the day" Time poverty associated with lower likelihood of fast food purchase" would benefit from being fleshed out a little for plausable explanations - one might intuitively think that fast food may be a faster option for those time poor.

Line 219 - 24 hour period? 4 a.m. to 3.59 a.m. rather than p.m.??

Line 228-229 reads a little clunky. Perhaps swap sentences around?

Line 296 - "...respondents that engaged in eating". Didn't they all eat something over 24 hours??

Line300 - It might be worth acknowledging in the Discussion that some practices are not likely to be done daily e.g. grocery shopping and future studies ...(where you mention future studies over weekly patterns).

(Apologies, I lost line numbering towards end of manuscript but where you cover Cluster 2: Home for Breakfast, Lunch and Dinner, change 56.9 to 57% for consistency.

Where you cover Cluster 5: Out for Dinner - Should the last sentence read 'lower among Hispanics (rather than non_Hispanics"??

Overall, a very thorough and interesting study with some valuable findings to guide future research and incorporate evidence from both food-practice and social practice into intervention design and delivery.  :)

Author Response

(The authors gave the same response as above.)

Round 2

Reviewer 1 Report

Thank you for revising your manuscript by considering my comments. Everything looks okay except the title.  Still, using the word "energy sustainability" looks inappropriate. As the authors wrote in Lines 445-470 (as the limitation), no findings are directly related to it. Also, I am curious why the energy-related transportation cost for food consumption wasn't fully discussed since the clusters were largely divided by locations. I think the authors may only focus on lifestyles and clusters which are supported by their analysis. 

Author Response

Reviewer 1

Thank you for revising your manuscript by considering my comments. Everything looks okay except the title.  Still, using the word "energy sustainability" looks inappropriate. As the authors wrote in Lines 445-470 (as the limitation), no findings are directly related to it.

We thank the reviewer for this comment, but we respectfully disagree that "implications for energy sustainability" should be removed from the title of our article. A core motivation for our work is exploring temporal related food consumption patterns as it relates to other consumptive activities, reflecting the focus of this Special Issue, which is “Public Health, Food and Environmental Policy Nexus in the Context of Civil Society.” We describe this connection to a specific consumption area, energy, in detail in the Background section of the manuscript, "2.3 Connections to energy use and sustainability" (lines 151 - 177). We have also added two sentences to the introduction that more clearly connect food consumption choices to energy sustainability. Additionally, our connection to energy sustainability is further reinforced by our choice of analytic method. Moreover, we employ a clustering method in our study that is often used to segment households by their daily electricity use patterns, and describe how this can be applied toward targeting of energy programs / interventions (lines 251 - 260). While we do not have specific findings on energy use in the home, as this data was not collected in our study, in the discussion we describe how the application of our food lifestyles approach has implications for efficiency, curtailment, and time-based electricity pricing intended to better align sources of renewable generation with demand, all related to sustainability issues (lines 445 - 470). In the conclusion we also discuss how food lifestyles can help understand electricity consumption changes at the grid level with implications for demand-side management and distributed energy resource deployment (lines 472 - 483). Given that energy sustainability is a core topic covered throughout the article -- in the background, methods, discussion and conclusions -- we feel that its placement in the title is important for describing its subject matter and helping food and energy audiences discover our work. All that said, we will defer to the Editor on the choice of title.

Also, I am curious why the energy-related transportation cost for food consumption wasn't fully discussed since the clusters were largely divided by locations. I think the authors may only focus on lifestyles and clusters which are supported by their analysis. 

We’re not sure if we fully understand this comment. However, we might expect “energy-related transportation costs” to be associated with a number of different factors, including 1) consumer travel for the explicit purpose of food consumption; 2) consumer travel (for other purposes) to locations where food is consumed; and 3) transportation associated with the manufacture and distribution of food products. The comment also implies that we might observe differences in energy-related transportation costs by location (home and not-home). To address this comment, we have added language to the Discussion about potential implications of our findings and future research related to food consumption habits and energy-related transportation costs.

Reviewer 2 Report

No further comments 

Author Response

Reviewer 2

No further comments 

We thank the reviewer for taking the time to review our revision.
